# Cardiac Repair and Clinical Outcomes of Stem Cell Therapy in Heart Failure: A Systematic Review and Meta-Analysis

**DOI:** 10.3390/diseases13050136

**Published:** 2025-04-29

**Authors:** Salman Muslem, Mariam AlTurani, Muhammad Bilal Maqsood, Maryam Al Qaseer

**Affiliations:** 1Salmaniya Medical Complex, Manama P.O. Box 11190, Bahrain; salman.a.muslim@gmail.com; 2Royal College of Surgeons, Ireland-Medical University of Bahrain, Manama P.O. Box 15503, Bahrain; maryam.masood.toorani@gmail.com; 3Clinical Excellence, Eastern Health Cluster, Dammam 32253, Saudi Arabia; maqsoodmub@gmail.com; 4Department of Cardiology, King Fahad Specialist Hospital, Eastern Health Cluster, Dammam 32253, Saudi Arabia

**Keywords:** mesenchymal stem cells, heart failure, ejection fraction, efficacy outcomes, quality of life

## Abstract

Background: While heart failure with reduced ejection fraction (HFrEF) remains a major global health burden, mesenchymal stem cell (MSC) therapy has emerged as a promising intervention designed to improve cardiac function and reduce morbidity among patients unresponsive to conventional treatments. MSC therapy has shown promise by targeting left ventricular pressure and improving wall thickness, contributing to reductions in HF-related morbidity and mortality rates. This systematic review and meta-analysis bridges a gap in current research through a focused examination of the most recent clinical trials to cohesively assess MSC therapy in HFrEF patients. Methods: We conducted a systematic review and meta-analysis of clinical trials published from 2018 onwards, which were obtained from multiple databases such as PUBMED, Scopus, EBSCO Medline, EBSCO CINAHL Science Direct, and the Cochrane Library. This review investigates the efficacy and safety outcomes of MSC therapy in patients above 18 years of age with a known diagnosis of heart failure with a reduced ejection fraction (HFrEF). The primary outcome was the change in the left ventricular ejection fraction (LVEF). Secondary outcomes encompassed several efficacy outcomes, such as Global Circumferential strain (GCS), the 6-Minute Walk Test (6MWT), Quality of Life (QoL), and major adverse cardiac events (MACE). A PRISMA flow diagram was constructed to illustrate the identification, screening, eligibility, and inclusion of studies at each stage of the review process. Results: A total of 330 studies were initially identified, but only 12 met the inclusion criteria. MSC therapy resulted in a small, non-significant improvement in LVEF (Hedges’ g = 0.096, *p* = 0.18) with low heterogeneity (I² = 0.5%). Only QoL showed significant improvement (Hedges’ g = −0.518, *p* = 0.01). No significant changes in other efficacy outcomes were observed. The therapy was not associated with an increased risk of MACE. Conclusion: While MSC therapy was safe and improved QoL for HFrEF patients, it did not significantly improve LVEF or other efficacy outcomes. Further large-scale, standardized trials are required to better understand the potential role of MSCs in heart failure (HF) therapy.

## 1. Introduction

### 1.1. Heart Failure with Reduced Ejection Fraction

Heart failure is a growing and significant problem in healthcare. Globally, the number of people affected with HF increased significantly from 25.4 million (95% UI 22.3–29.2) in 1990 to over 55.5 million (95% UI 49.0–63.8) in 2021 [1]. Furthermore, it is a leading cause of hospitalization and an economic burden. HF refers to a complex clinical syndrome where inadequate pumping of blood from the heart leads to the hypoperfusion of other organs and tissues [2]. The pathophysiology of HF is attributed to different underlying causes that strain the heart, predominantly coronary artery disease, in addition to other inflammatory and infectious pathologies [2]. There are different phenotypes of HF, including heart failure with a reduced ejection fraction (HFrEF) and heart failure with a preserved ejection fraction (HFpEF). By definition, HFrEF is the term given to patients with an ejection fraction below 40%, while heart failure with a mildly reduced ejection fraction is defined as ejection fraction between 41% and 49%. The term HFpEF refers to patients with an ejection fraction that lies above 50% [3]. The left ventricular ejection fraction (LVEF) is the primary measure of efficacy explored in this review. It is defined as the proportion of blood pumped from the left ventricle during systole to the body, and it reflects overall cardiac performance. In the HFrEF cohort, this value is reduced, and improvement can be seen through an increase in the LVEF percentage following treatment. This review solely examines HFrEF patients and their physiological abnormalities, which include the enlargement and weakening of the left ventricle, precipitating systolic failure.

Guideline-directed medical therapy (GDMT) is the mainstay pharmacological therapy for patients with HFrEF. It includes four main drug classes: renin–angiotensin system inhibitors, mineralocorticoid receptor antagonists, cardioselective beta blockers, and sodium-glucose cotransporter 2 inhibitors (SGLT2i) [4,5]. There has been a paradigm shift in how heart failure is treated following the introduction of the angiotensin receptor/neprilysin inhibitors and SGLT2i. Today, these four pillars of therapy have significantly improved patient outcomes, reducing both hospitalizations and mortality. However, despite these advancements, there remains a residual risk that propels patients along the HF trajectory. A small subset of this cohort develops advanced HF, leaving them refractory to the conventional four pillars of therapy. Consequently, the five-year mortality rate for heart failure has remained largely unchanged over the past two decades [6]. It has been postulated that, for this subset of patients, further improvements cannot be achieved through neurohormonal therapies alone. Rather, they will need to address Laplace’s therapeutics, which revolve around reducing wall stress. This can be accomplished by reducing the radius of the left ventricle, improving wall thickness, or lowering pressure. The current therapies offered for such patients include mechanical circulatory support and heart transplant. More recently, mesenchymal stem cell therapy has shown promise by targeting left-ventricular pressure and improving wall thickness, contributing to reductions in HF-related morbidity and mortality rates [7].

### 1.2. Mesenchymal Stem Cells (MSCs)

Stem cells are a special group of cells that have the ability to self-renew and differentiate along various cell lineages [8]. These cells can be subclassified into different groups according to the area from which they are derived as well as their level of potency. Totipotent stem cells, derived from blastocysts, can differentiate into cells from any of the three germ layers and into extraembryonic cell types. On the other hand, human embryonic stem cells are termed pluripotent as they possess the ability to differentiate into any mature cell from the three germ layers, except for extraembryonic cells. Embryonic stem cells have substantial potential in the field of regenerative medicine, but their clinical application is limited by the degree of ethical implications associated with their use [9]. Embryonic stem cells further mature into adult stem cells (ASCs), which are limited in terms of their differentiation pathways. These cells are termed multipotent.

Mesenchymal stem cells (MSCs) are a group of multipotent ASCs found in various tissues, including bone marrow, adipose tissue, umbilical cord tissue and blood, amniotic fluid, menstrual blood, dental pulp, testes, dermis, spleen, pancreas, and lung [10]. The International Society for Cellular Therapy has proposed minimum criteria for defining MSCs. These cells must (a) possess a specific set of cell surface markers, i.e., cluster of differentiation CD73, D90, and CD10 and lack expression of CD14, CD34, CD45, and human leukocyte antigen-DR; (b) exhibit plastic adherence; and (c) can differentiate in vitro into adipocytes, chondrocytes, and osteoblasts [11]. These cells are easily accessible, isolatable, and expandable due to their genomic stability. Additionally, they demonstrate a remarkable migratory capacity and high expansion rates and can avoid allogeneic responses after transplantation. Moreover, there are relatively fewer ethical implications regarding their use in comparison to embryonal and other forms of stem cells [12].

### 1.3. Improving LVEF and Secondary Outcomes

On a cellular level, chronic HF is characterized by chronic, low-grade inflammation; reduced vasculogenesis; increased apoptosis; and progressive interstitial deposition of collagen. While most bone marrow-derived stem cells (BM-MSCs) exhibit their effects by reducing inflammatory damage, MSCs have also been shown to possess the greatest potential to drive cardiomyocyte proliferation and regeneration [13]. MSC therapy in HF is primarily delivered through intracoronary, intravenous, or intramyocardial routes via autologous or allogeneic cells. These cells mainly exert effects through paracrine mechanisms, which include protective and reparative mechanisms. Anti-apoptotic action is exerted on injured cardiomyocytes due to the transfer of healthy mitochondria from the MSCs via tunneling nanotubes and the secretion of exosomes carrying regenerative signals. MSCs are seen to activate survival pathways under low-oxygen conditions, strengthening their therapeutic potential in heart failure treatment. More recent studies have shown that angiogenesis is a key mechanism in improving cardiac function. Research has demonstrated MSCs’ ability to improve clinical, functional, structural, and biochemical markers regarding HF. The MSC-HF trial, which utilized intra-myocardial injections of autologous MSCs, showed that MSCs were associated with an improvement in the left ventricular ejection fraction (LVEF) and a reduction in hospitalization rates and confirmed their safety [14]. These findings were confirmed by Bartolucci et al.’s study, wherein the employment of umbilical-cord-derived MSCs improved both left ventricular function and quality of life (QoL) [15].

In the studies included in this review, efficacy was commonly assessed through changes in LVEF, 6MWT, quality of life indices and biomarker changes in NT-proBNP levels. Although cardiomyocyte proliferation is considered a potential mechanism of regeneration, it was not measured as a direct outcome in the clinical trials examined.

### 1.4. The Importance of This Review

Heart failure imposes a substantial financial burden worldwide, with the global cost of care expected to increase 2.5-fold by 2030, according to the American Heart Association (AHA) [16]. The current treatment modalities, which include pharmacological therapies, device-based interventions, and lifestyle changes, have helped to reduce cardiac morbidity for some patients. However, it is important to accentuate the importance of research and reviews to investigate innovative therapies that restore function and reverse cardiac damage as their foremost principle [17]. Although other systematic reviews on this topic have been published, this review specifically focuses on MSCs and examines only the most recent trials. Existing systematic reviews either include outdated studies or fail to focus solely on HFrEF patients, thereby limiting their clinical relevance. Consequently, this systematic review and meta-analysis bridges this gap by recruiting the most recent clinical trials to cohesively assess MSC therapy in HFrEF patients through an updated and focused approach. This systematic review and meta-analysis compiles and evaluates the latest trials examining the efficacy and safety of MSC therapy for HFrEF patients. It is hypothesized that the administration of MSCs is both safe and efficacious in the treatment of HFrEF.

## 2. Materials and Methods

### 2.1. Protocol Registration

In line with the Preferred Reporting Items for Systematic Reviews and Meta-Analyses (PRISMA) guidelines [18], this study was registered with the International Prospective Register of Systematic Reviews (PROSPERO). The registered review can be found using the PROSPERO ID CRD42023439429.

### 2.2. Inclusion and Exclusion Criteria

Interventional studies were chosen for this review. These encompassed randomized control trials (RCTs), single-arm open-label studies, and non-randomized clinical trials. Only studies published in English (to ensure global generalizability) from 2018 onwards were included to reflect the most current advancements in stem cell treatment, including refinements in cell sourcing, delivery methods, and standardized outcome reporting, and to ensure accurate and consistent data extraction. Regarding the duration of follow-up, the studies had to include multiple endpoints with a minimum of 30 days of follow-up. The values obtained for this study were primarily derived from the one-year follow-up endpoint; however, some studies did not have a one-year follow-up period. The studies included were conducted in hospital settings, including tertiary cardiac centers, which facilitated masking and blinding. However, it is important to note that the actual implementation of these protocols may vary across studies according to local hospital practices and guidelines. The study population for this review included HFrEF patients aged 18 and above, inclusive of all genders, races, and ethnicities. The participants had to have a confirmed diagnosis of HFrEF made by a cardiologist according to AHA guidelines. HF could have resulted from primary or secondary cardiomyopathy. Exclusion criteria included pediatric patients and individuals with heart failure with preserved ejection fraction (HFpEF).

The included studies had to evaluate young patients treated with MSCs as the intervention group, with no restrictions on the route, dose, duration, or frequency of the intervention. One study [19] looked at cardiosphere-derived stem cells (CSCs), which exert several overlapping immunomodulatory effects and have a regenerative potential similar to that of MSCs [20]. In this review, the outcome measures were categorized as either efficacy or safety outcomes. Moreover, the studies were synthesized and grouped based on the PICO framework elements, which encompass population, intervention, comparison, and outcomes to facilitate the evaluation of evidence in the studies. This process also incorporated the eligible studies’ designs. The population consisted of adult patients with HFrEF, the intervention was MSC treatment, the control could be any other standard treatment or placebo, and the outcomes were divided into a primary outcome (LVEF %) and various secondary outcomes as mentioned in Section 2.2.1 and Section 2.2.2.

Regarding the minimal methodological quality requirement, only studies that demonstrated a low to moderate risk of bias based on the Cochrane Risk of Bias (RoB 2) tool were included. Additionally, all the included studies had to utilize a clearly defined intervention involving MSCs, provide quantitative data for at least one pre-specified primary or secondary outcome, and use appropriate control groups (except for single-arm interventional studies with a justified methodology).

#### 2.2.1. Primary Outcome

The primary outcome of this study is the change in LVEF (%), which is a measure of efficacy. LVEF (%) can be determined through different non-invasive modalities, including echocardiography, magnetic resonance imaging (MRI), and computed tomography [21], and it serves as a reliable prognostic tool that predicts major adverse cardiac events (MACE) and cardiac mortality. Therefore, LVEF is commonly used to predict responses to HF therapies.

#### 2.2.2. Secondary Outcomes

The secondary outcomes of this study are divided into various continuous efficacy outcomes and a single dichotomous safety outcome. The secondary efficacy outcomes are subdivided into the categories described below.

Measure of cardiac function:
○Global circumferential strain (GCS)(%): This serves as a prognostic indicator in myocardial disease, particularly when associated with fibrosis [22].Functional Capacity: These outcomes are relevant since this particular cohort of patients suffers from a disease that predisposes them to exercise intolerance [23]. The tests used to assess these outcomes are described below. ○The Minnesota Living with Heart Failure Questionnaire (MLHFQ) is a subjective questionnaire that evaluates the physical and emotional impacts of heart failure on patients [24].○The 6-Minute Walk Test (6 MWT) (m) is an inexpensive and easily performed test that has been widely applied in studies that assess HF interventions [25].Structural: These outcomes reflect left ventricular remodeling patterns and are independent predictors of survival for HF patients [26].○Left ventricular end-diastolic volume index (LVEDVI) (mL/m^2^).○Left ventricular end-systolic volume index (LVESVI) (mL/m^2^).Morphological: A reduction in these outcomes is indicative of less adverse cardiac remodeling and hence reduced wall stress, thereby addressing Laplace’s therapeutics [27].○Sphericity (mL).○Scar size (%).○Scar tissue mass (g).Biochemical markers:○N-terminal prohormone brain natriuretic peptide (NT-proBNP) (pg/mL) is a cleaved form of proBNP that is secreted into the ventricles in response to volume expansion and pressure overload. This biomarker provides insight into the severity of disease progression [28,29,30].

The single safety outcome assessed in this review was the incidence of MACE, including cardiac death, arrhythmias, myocardial infarction, stroke, heart failure hospitalization, revascularization procedures, and other events [31].

### 2.3. Identification of Studies

Electronic searches were performed in multiple databases, including PubMed, Scopus, EBSCO Medline, EBSCO CINAHL, the Cochrane Library, and ScienceDirect. The following additional filters were used: open access, English language, years 2018–2023, and clinical trials. No other resources were utilized aside. The searches were performed on 29 November 2023, and the mesh terms used for each database are included in Table 1.

### 2.4. Selection of Studies

The studies were selected using the Covidence systematic review tool. Two reviewers performed the initial title and abstract screening in a blinded fashion; conflicts were resolved by a third reviewer. Afterwards, full-text screening was performed similarly. Specifically, the two reviews were blinded during the screening process, and any discrepancies were resolved by a third reviewer. The screening process was strictly based on the inclusion criteria and completed independently by reviewer 1 and reviewer 2 to reduce bias. Studies that were deemed ineligible were excluded in the screening process.

Qualitative data extraction involved obtaining baseline characteristics on the patient population, interventions, control groups, and other methodological factors such as the types of MSCs. Other important information was derived from the discussion, limitations, and conclusions sections of the studies. Qualitative data extraction was carried out by reviewer 1, while quantitative data extraction was conducted by reviewer 2. For the quantitative data, data extraction was subdivided into continuous and binary outcomes. Standard deviations and means were recorded for all the continuous outcome measures. The binary outcome measure, MACE, was measured by extracting the number (N) and rate (%). Subsequently, forest plots were made accordingly for all the individual outcomes to demonstrate the treatment effect ascertained by the reviewers.

### 2.5. Risk Assessment

We conducted a quality assessment and determined the risk of bias using the Cochrane risk-of-bias tool for randomized trials (RoB 2). Each of the included studies was assessed for potential methodological flaws via domains such as the risk of bias arising from the randomization process, deviations from the intended interventions, missing outcome data, measurement of the outcome, and selection of the reported results. Bias was assessed using the judgments (high, low, or unclear) made independently by two authors, and disagreements were resolved through a consensus [32].

### 2.6. Measures of Treatment Effect

For continuous outcomes, the mean difference was measured. The mean differences across the studies were standardized using Hedges’ g measure. The effect size was quantified using value cutoffs that could either be positive or negative and interpreted as small, medium, or large effects, represented as 0.2, 0.5, and 0.8, respectively [33]. The further the value from zero and the closer to 1.0 in the positive direction, the stronger the effect in favor of the intervention. Conversely, the closer the value is to being negative, the greater the effect in favor of the control. A value of zero indicates there is no difference between the groups and thus no treatment effect. MACE constituted the only binary outcome. Hence, this was measured through a log risk ratio. Values were either positive, negative, or zero. A positive value means that the outcome was in favor of the treatment group, while a negative value indicates that the treatment effect was in favor of the control group. A value of zero indicates that there was no treatment effect [34].

### 2.7. Assessment of Heterogeneity

Heterogeneity was assessed using the I2 score, demonstrated as a percentage. The closer the score to 0%, the lower the heterogeneity across the studies, with 0% indicating no heterogeneity was observed. A *p*-value obtained from the chi-squared heterogeneity test was used alongside each I2 score to determine whether a score was statistically significant [35].

## 3. Results

### 3.1. Description of the Search

After the initial search on multiple databases, the search hits resulted in 330 references; these were exported to the Covidence systematic review tool [36]. A total of 77 duplicates were removed from the references. In the title and abstract screening stage, 253 studies were evaluated, resulting in the exclusion of 211 studies. Subsequently, 42 studies underwent full-text assessment for eligibility, which resulted in the exclusion of 30 studies. The reasons for exclusion were as follows: no results reported (four studies), wrong outcomes provided (two studies), wrong comparator used (one study), wrong patient population assessed (four studies), wrong intervention employed (four studies), and wrong study design used (fifteen studies). Consequently, a total of 12 studies were eligible for this review. A summary of the search results demonstrated through a PRISMA flow diagram is given in Figure 1.

### 3.2. Study Characteristics

The 12 eligible studies underwent qualitative and quantitative data extraction. These studies were available as abstracts and full texts and published in English. The qualitative data are shown in Table 2. Throughout this review, the studies will be referred to by their corresponding numbers assigned owing to their relevance asshown in Table 2.

Except for Studies 3 and 12, [46,47] which were single-arm, open-label, interventional studies, all the remaining studies were RCTs. With regard to the RCTs, phase I, II, and III trials were involved. A phase I RCT was conducted in Study 2 [37], whereas Studies 1, 8, 10, and 11 [38,39,40,41] were phase II RCTs. Additionally, Study 9 [42] was a phase III RCT. As for masking, quadruple masking was performed in Studies 1 and 2 [37,38], while the other studies (Studies 4, 5, 6, 7, 8, 9, 10, and 11 [19,39,40,41,42,43,44,45]) were double-blind, placebo-controlled studies. Regarding the study settings, Studies 7, 10, and 11 [19,40,41] were multicenter studies, while Study 5 [44] was a single-center study that took place in Nanjing, China. Studies 4, 8, and 11 were based on European populations [39,41,43]. Studies 1 and 2 were conducted in the United States [37,38]. Study 12 [46] was conducted in Japan. In terms of follow-up periods, the majority of the studies, i.e., 1, 2, 5, 6, 7, 9, 10, and 11 [19,37,38,40,41,42,44,45], had follow-up values for 12 months. However, Studies 3 and 8 [39,47] followed up with patients for four years, while Study 4 had a follow-up of 30 days.

Regarding the population size, most of the studies (Studies 2, 3, 4, 5, 6, 8, 10, and 12 [37,39,40,43,44,45,46,47]) had fewer than 100 participants, while others had populations that ranged from 100 to 565 participants (Studies 1, 7, 9, and 11 [19,38,41,42]). Regarding the patient population characteristics, all the patients were adults, ranging from a mean age of 53.5 to 70 years old. Another important characteristic amongst the studies was the LVEF of the patients at baseline. The studies included in this review set different baseline LVEF cutoffs for patient enrollment. Study 3 [47] had the strictest requirement, with an LVEF of less than 35%, while Studies 1, 5, 6, 7, and 9 [19,38,44,45] included patients with an LVEF below 40%. Additionally, Studies 3, 8, 10, 11, and 12 [39,40,41,46,47] accepted patients with an LVEF under 45%. Furthermore, Studies 1, 2, and 3 [37,38,47] had patients who were on optimal GDMT. The mean patient NYHA class was also reported for most of the studies, and the values can be seen in Table 2.

Various routes for stem cell administration were utilized in the studies. Studies 1, 4, and 9 [38,42,43] included patients who underwent transendocardial injection of MSCs. Intramyocardial injection of stem cells was performed in Studies 2, 5, 8, and 10 [37,39,40,44]. The authors of Study 4 [43] employed stem cell implantation in the intervention group. Finally, an intracoronary infusion of stem cells was performed in Studies 6, 7, and 12 [19,45,46]. Concerning the subtypes of stem cells, autologous stem cells were administered in patients in Studies 1, 3, 4, 6, 8, and 12 [38,39,43,45,46,47], while allogeneic stem cells were used in Studies 2, 5, 7, 9, 10, and 11 [19,37,40,41,42,44]. Additionally, different sources of MSCs were studied. Specifically, BM-MSCs were used in Studies 1, 4, 6, 8, 9, and 12 [38,39,42,43,45,46]. In contrast, skeletal stem cells were used in Study 3 [47], while umbilical-cord-derived MSCs were utilized in Study 5 [44], and cardiosphere-derived cells were given to patients in Study 7 [19]. Finally, adipose-tissue-derived MSCs were employed in Studies 10 and 11 [40,41]. For the control arm, placebos were administered to the control group in the majority of the studies, such as Studies 1, 2, 7, 8, 10, and 11 [19,37,38,39,40,41]. Open-label single-arm interventional studies, such as Studies 3 and 12 [46,47], had no control group. A sham procedure was performed in Studies 4 and 9 [42,43]. Finally, the control group in Study 5 [44] underwent a coronary artery bypass graft alone.

### 3.3. Risk of Bias in the Included Studies

The RoB 2 tool was used to evaluate bias that may have influenced the results of the 12 selected studies. The tool assessed five major domains of bias, as mentioned previously in Section 3.4. Overall, there was a moderate to low risk of bias across all the included studies. The risk of bias for each study is demonstrated through the legend in Figure 2.

### 3.4. Synthesis of the Results

#### 3.4.1. Primary Outcome: LVEF

Due to the discrepancies in sample sizes and effects, the Hedge’s g statistical method was used to calculate the effect size. The meta-analysis for LVEF resulted in a Hedges’ g = 0.096 (95% CI: −0.045 to 0.237; *p* = 0.18), as shown in Figure 3. Theoretically, the larger the value, the greater the effect size. The results showed a value below 0.2, signifying that the treatment effect size was small. Additionally, the effect was not statistically significant, as shown by the *p*-value. Specifically, small treatment effect sizes like this do not favor the objectives of this review, as a clinically significant improvement for patients with HFrEF should align with large effect sizes. The heterogeneity across the studies was low, as demonstrated by the I^2^ = 0.5%. This indicates that the findings were consistent across the different studies. Hence, the statistical analyses show that the MSC interventions were not associated with significant LVEF improvements in the HFrEF cohort.

#### 3.4.2. Secondary Outcomes: Efficacy

The efficacy of MSCs was also assessed through secondary outcomes. Another element of the functional assessment of HFrEF patients was GCS. The two studies [37,38] that assessed GCS as an outcome measure indicated it had a negative and small treatment effect (Hedges’ g = −0.372; 95% CI −0.898 to 0.153, *p* = 0.16). This finding indicates that the control group outperformed the intervention group. That said, a low heterogeneity was observed across the two studies (I^2^ = 0%), which can be seen in Figure 4. This particular trend was also seen for the other secondary efficacy outcomes. In regard to structural outcomes, the results were as follows: Hedges’ g = −0.249 (95% CI: −0.737 to 0.240, *p* = 0.35), Hedges g’ = −0.249 (95% CI: −0.737 to 0.240, *p* = 0.32), and Hedges’ g = −0.056 (95% CI: −0.544 to 0.431, *p* = 0.82) for LVEDVI (mL/m^2^), LVESVI (mL/m^2^), and sphericity (mL), respectively; these are visualized in Figure 5, Figure 6 and Figure 7. As for the assessment of morphological changes, scar size (%) [19,37,38,44] had a Hedges’ g = −0.125 (95% CI: −0.411 to 0.160, *p* = 0.39), while scar tissue mass (g) [19,37,38] had a Hedges’ g = −0.193 (95% CI: −0.566 to 0.179, *p* = 0.31). However, the results showed low heterogeneity across the studies, with an I^2^ of 19.3% and a *p*-value of 0.236, indicating no significant variation. These findings for scar size and scar tissue mass are shown in Figure 8 and Figure 9, respectively. Similar findings were obtained regarding outcome measures that assessed functional capacity, with a notable significance in heterogeneity in the 6MWT (m) [37,38,40], with Hedges’ g = −0.131 (95% CI: −1.336 to 1.074, *p* = 0.83) (I^2^ = 92%, *p* = 0.00) [48]. The results of the 6MWT are depicted in Figure 10.

In contrast to what was observed for the majority of the variables measured in this review, the MSC group demonstrated a significant change in the MLHFQ summary score [37,38,44], with a value of Hedges’ g = −0.518 (95% CI: −0.919 to −0.118, *p* = 0.01) relative to the placebo group. These results were shown to be homogenous across the studies (I^2^ = 0%, *p* = 0.626), as seen in Figure 11. Finally, biochemical changes were assessed through the N-terminal prohormone of brain natriuretic peptide (NT-proBNP) (pg/mL) [37,38,41], with findings that were similar to those for the other outcomes, featuring a small treatment effect with low heterogeneity (Hedges’ g = −0.130; 95% CI: −0.424 to 0.165, *p* = 0.39) (I^2^ = 0%, *p* = 0.762), which can be found in Figure 12.

#### 3.4.3. Secondary Outcome: Safety/MACE

The two studies [19,38] that assessed the safety of MSCs in HFrEF patients demonstrated an insignificant treatment effect in the intervention group when compared to the control group. A random-effects model was utilized, and a log risk ratio was obtained for the studies that reported MACE, which had an overall effect of RR = 0.044 (95% CI: −0.066 to 0.155, *p* = 0.432) (I^2^ = 0%). Hence, the results were consistent as the heterogeneity was considerably low, as shown in Figure 13 [34].

### 3.5. Meta-Analysis Findings

A detailed visual representation of all the forest plots of the measured variables based on the meta-analysis is given in the figures below.

#### 3.5.1. Statistical Analyses of Continuous Outcomes

A random effects model was used for LVEF assessment. There was no significant difference between the two groups in terms of LVEF changes (Hedges’ g = 0.096, 95% CI: −0.045 to 0.237, *p* = 0.18). The heterogeneity across the studies was low, as demonstrated by the I^2^ = 0.5% (*p* = 0.65). This indicates that the findings were consistent across the different studies.

**Figure 3 diseases-13-00136-f003:**
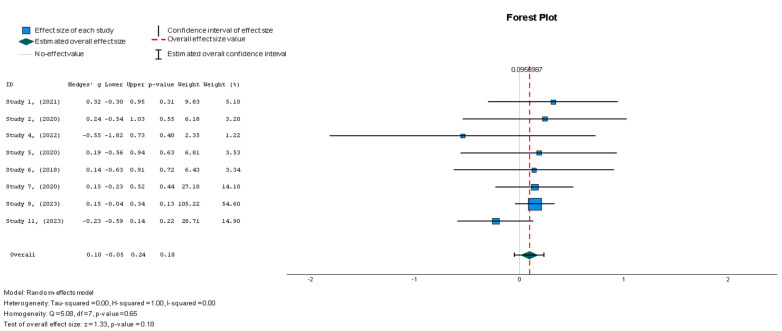
Left ventricular ejection fraction—LVEF (%) [19,37,38,41,42,43,44,45].

The results conveyed a statistically insignificant decrease in GCS compared to the control group in the patients who underwent stem cell therapy (Hedges’ g = −0.372, 95% CI: −0.898 to 0.153, *p* = 0.16). There was low heterogeneity, with an I^2^ = 0% (*p* = 0.80).

**Figure 4 diseases-13-00136-f004:**
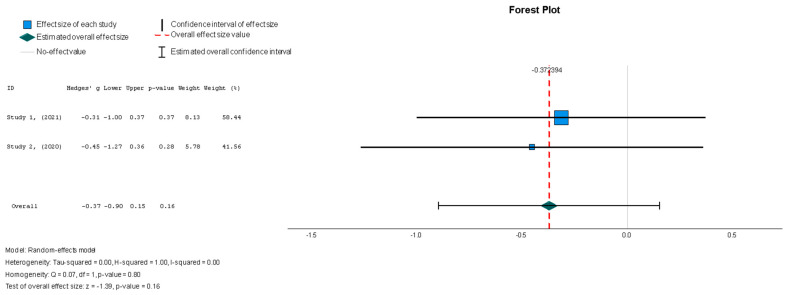
Global circumferential strain—GCS (%) [37,38].

Stem cell therapy was associated with a statistically insignificant reduction in the LVEDVI, as indicated by the results (Hedges’ g = −0.249, 95% CI: −0.737 to 0.240, *p* = 0.35). These results were also supported by low heterogeneity (I^2^ = 0%, *p* = 0.78).

**Figure 5 diseases-13-00136-f005:**
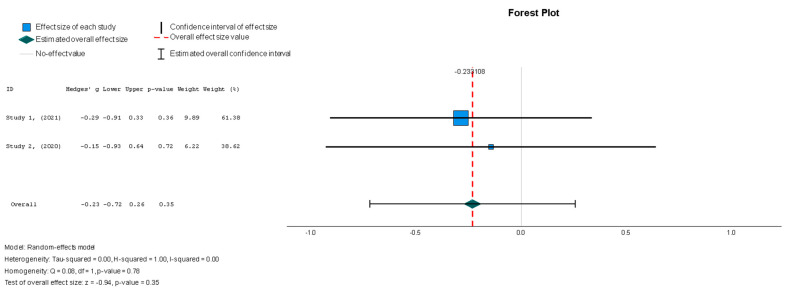
Left ventricular end-diastolic volume index—LVEDVI (mL/m^2^) [37,38].

Stem cell therapy resulted in a statistically insignificant decrease in the LVESVI u (Hedges g’ = −0.249, 95% CI: −0.737 to 0.240, *p* = 0.32). There was no significant heterogeneity across the studies (I^2^ = 0%, *p* = 0.80).

**Figure 6 diseases-13-00136-f006:**
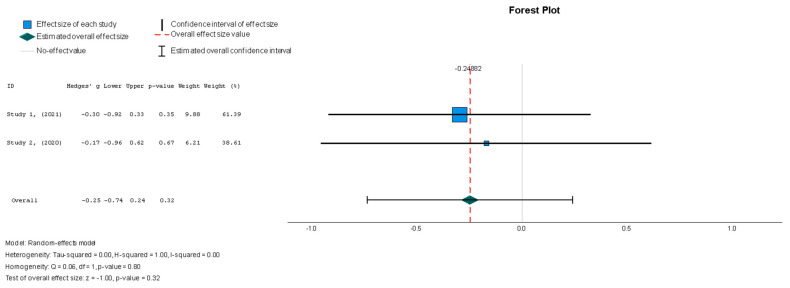
Left ventricular end-systolic volume index—LVESVI (mL/m^2^) [37,38].

Stem cell therapy had a small effect on sphericity without displaying statistical significance (Hedges’ g = −0.056, 95%: −0.544 to 0.431, *p* = 0.82). No significant heterogeneity was seen across the studies (I^2^ = 0%, *p* = 0.446).

**Figure 7 diseases-13-00136-f007:**
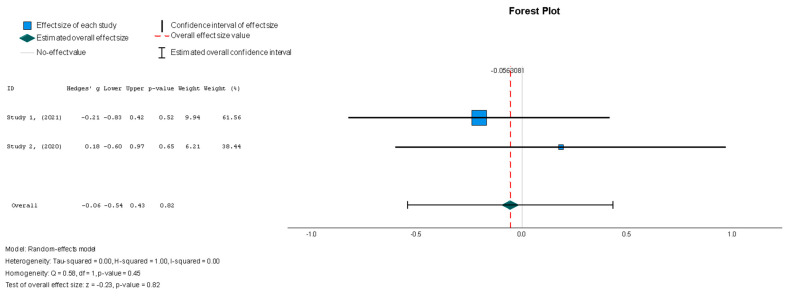
Sphericity (mL) [37,38].

When assessing the effect of stem cell therapy on scar size, there was a weak effect with no statistical significance (Hedges’ g = −0.125, 95% CI: −0.411 to 0.160, *p* = 0.39). These findings were homogenous but not statistically significant (I^2^ = 0%, *p* = 0.357).

**Figure 8 diseases-13-00136-f008:**
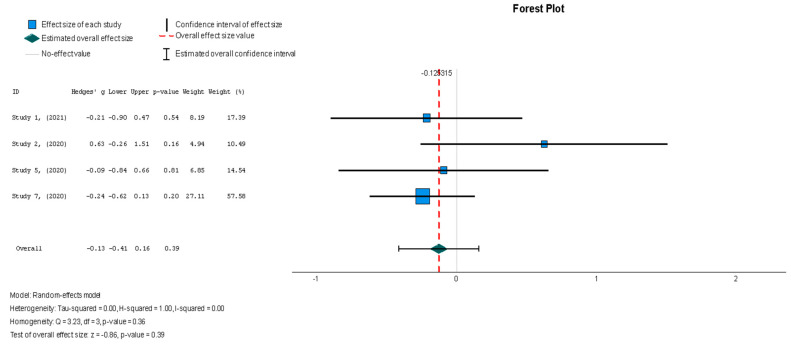
Scar size (%) [19,37,38,44].

Following the trend of the previous variables, stem cell therapy failed to demonstrate a significant difference in scar tissue mass when compared with the placebo (Hedges’ g = −0.193, 95% CI: −0.566 to 0.179, *p* = 0.31). The results were non-significantly homogenous across the different studies (I^2^ = 19.3%, *p* = 0.236).

**Figure 9 diseases-13-00136-f009:**
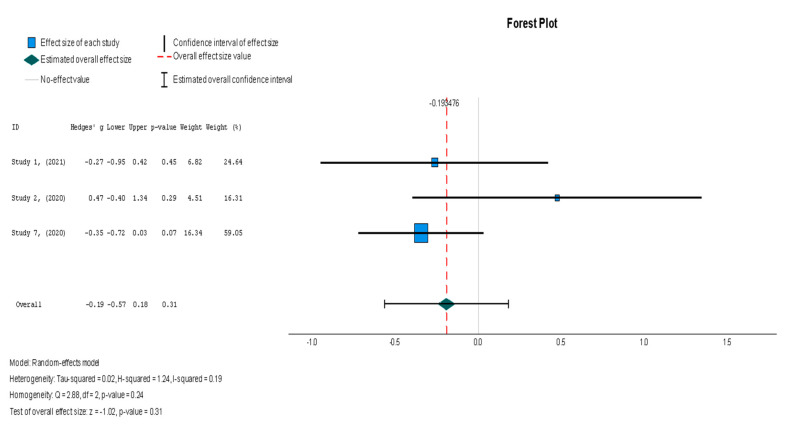
Scar tissue mass (g) [19,37,38].

No significant change in the 6MWT scores was observed when comparing stem cell therapy to the placebo (Hedges’ g = −0.131, 95% CI: −1.336 to 1.074, *p* = 0.83). Moreover, the selected studies exhibited significant heterogeneity (I^2^ = 92%, *p* = 0.00).

**Figure 10 diseases-13-00136-f010:**
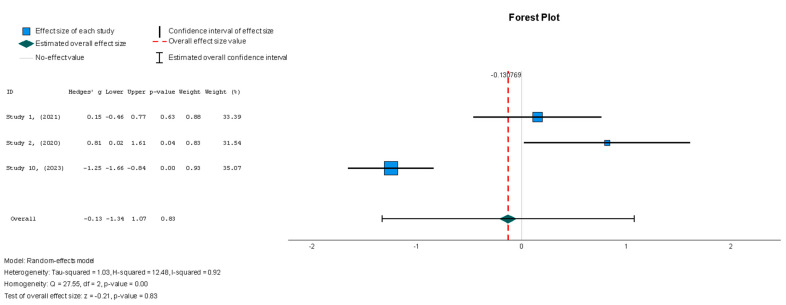
Six-minute walk test—6MWT (m) [37,38,40].

In contrast to the majority of the variables measured in this review, the stem cell intervention group demonstrated a significant change in the MLHFQ summary score (Hedges’ g = −0.518, 95% CI: −0.919 to −0.118, *p* = 0.01) relative to the placebo group. These results were shown to be homogenous across studies; however, the homogeneity was insignificant (I^2^ = 0%, *p* = 0.626).

**Figure 11 diseases-13-00136-f011:**
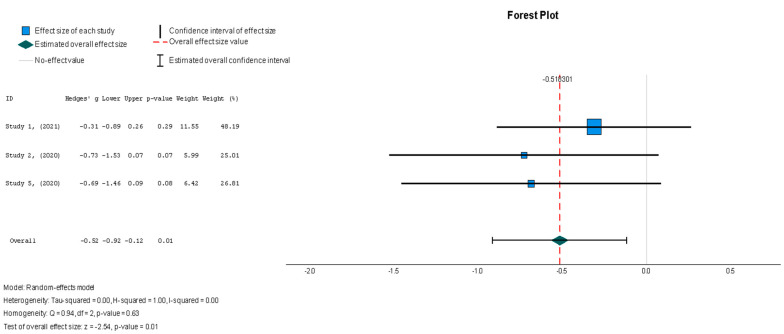
Minnesota Living with Heart Failure Questionnaire (MLHFQ) summary score [37,38,44].

The stem cell group did not have significantly decreased NT-proBNP levels when compared to the placebo group (Hedges’ g = −0.130, 95% CI: −0.424 to 0.165, *p* = 0.39). The results were insignificantly homogenous (I^2^ = 0%, *p* = 0.762).

**Figure 12 diseases-13-00136-f012:**
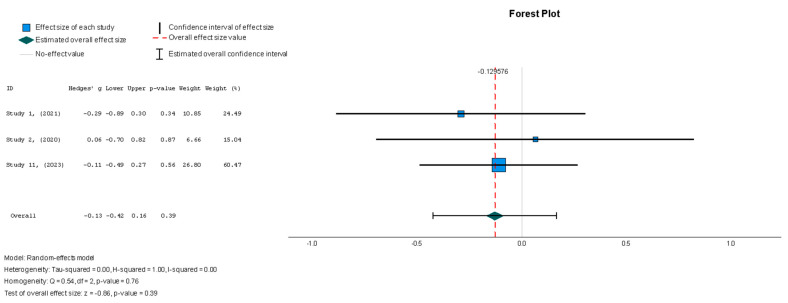
N-terminal prohormone of brain natriuretic peptide—NT-proBNP (pg/mL) [37,38,41].

#### 3.5.2. Statistical Analyses of Binary Outcomes

In terms of the safety outcome, MSC administration was not associated with a significant increase in MACE, as the results from Studies 1 and 7 showed no significant differences between the MSC and control groups (OR = 0.51, *p* = 0.25, 95% CI = −0.37 to 1.39).

**Figure 13 diseases-13-00136-f013:**
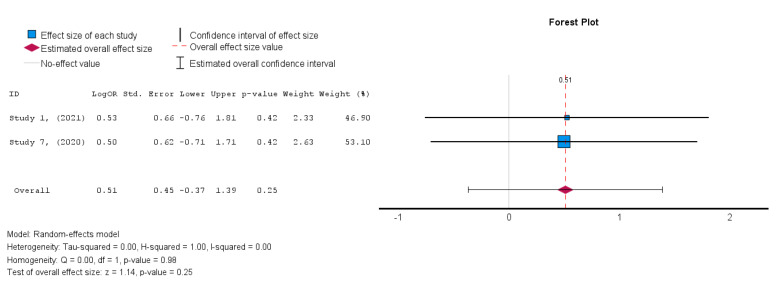
Major adverse cardiac events (MACE) [19,38].

## 4. Discussion

### 4.1. Summary of the Main Results

Overall, the results indicated that subjecting HFrEF patients to MSC therapy did not produce statistically significant improvements in LVEF (%) or MACE. Hence, the statistical values derived from the meta-analysis were inadequate for drawing conclusions. With regard to LVEF (%), a small treatment effect size was seen, with a *p* value > 0.05. Hence, MSCs’ effect on the primary outcome was neither clinically nor statistically significant. As for the secondary efficacy outcomes, negative Hedges G values with low heterogeneity were found, implying that greater change was consistently observed in the control group in comparison to the intervention group. Lastly, no statistically or clinically significant changes in MACE were reported when comparing the intervention groups to the controls. Conversely, there was a statistically significant improvement in QoL, as measured by a subjective increase in the MLHFQ score. This may indicate that stem cell therapy is associated with symptomatic improvement in the absence of LVEF, structural, and morphological changes, raising the question of whether the intervention may be more suitable for HFpEF patients, who display symptoms in the presence of a preserved ejection fraction.

### 4.2. Agreements and Disagreements with Other Studies

When comparing this review to other existing studies and reviews, a discrepancy in the conclusions is prominent. For instance, Jayaraj JS et al. found that stem cell therapy significantly enhanced both safety and efficacy. The results, depicted as the weighted mean difference, showed notable changes in LVEF (%) and LVESVI (mL/m^2^), with improvements of 4.58% (95% CI: 3.73 to 5.43, *p* = 0.00001) and a reduction of −5.18 mL (95% CI: −9.74 to −0.63 mL, *p* = 0.01), respectively [49]. Similarly, Kalou et al. showed that BM-MSCs were associated with statistically significant improvements in 6MWT (m) scores, increasing by 27.86 m (95% CI 0.11 to 55.6), and LVEF, increasing by 6.37% (95% CI 5.48 to 7.26), supporting the intervention’s efficacy [50]. Another review published in 2015 that analyzed 48 studies also found that BM-MSCs were associated with a change in LVEF, this time amounting to 2.92% (95% CI 1.91 to 3.92, *p* < 0.00001), which was statistically significant but clinically insignificant [51]. Several other systematic reviews on stem cell efficacy in HF have also reported significant improvements in LVEF (%) [50,52,53,54].

A crucial biological explanation for why MSCs failed to produce a significant improvement in LVEF in the current review is the low cellular retention rate associated with stem cell engraftment onto cardiac tissue. This has been observed with intracoronary and intramyocardial administration methods in particular. For instance, preclinical studies have shown that intramyocardial injections of MSCs resulted in negligible cellular retention in the heart after 24 h since administration. In the studies included in this review, minimal measures were undertaken to maximize cellular retention. Therefore, the low retention rates could limit the promising anti-fibrotic and anti-inflammatory effects associated with MSCs [55].

In addition to low cellular retention, another contributing factor is the short-lived nature of MSC paracrine signaling. Various trials, including the BOOST trial, have demonstrated that the therapeutic benefits of MSCs tend to diminish beyond six months. This decline has been attributed to the fact that undifferentiated MSCs lose their regenerative secretory profile as they begin to differentiate, preventing them from producing a sustained therapeutic benefit [56].

MSCs were also initially thought to regenerate damaged myocardium by interacting with c-kit+ cells, which were believed to be cardiac stem cells capable of differentiating into mature cardiomyocytes. However, recent lineage-tracing studies have shown that c-kit+ cells primarily differentiate into endothelial cells, not cardiomyocytes. Most new cardiac muscle cells are now thought to originate from pre-existing cardiomyocytes through limited self-replication. Therefore, the concept that MSCs induce cardiac regeneration by expanding an endogenous cardiac stem cell population and increasing its commitment towards a cardiomyocyte lineage is no longer supported. Instead, this evidence supports the notion that angiogenesis may be the primary mechanism of MSC-induced myocardial repair [57].

Supporting this, studies have also shown that MSCs cultured under normal oxygen conditions do not significantly express angiogenic factors like vascular endothelial growth factor (VEGF). Preconditioning MSCs in low-oxygen conditions in vitro more closely mimics their natural hypoxic environment, enhancing the expression of angiogenic factors such as fibroblast growth factor and VEGF. Additionally, MSCs exposed to inflammatory factors such as tumor growth factor alpha also increase VEGF expression. Strategies aimed at enhancing cellular retention and pre-conditioning MSCs to optimize the expression of angiogenic and anti-fibrotic factors have been developed [58].

The observed improvement in MLHFQ despite the absence of improvement in other variables remains unexplained. One possible explanation is the immunomodulatory effect of MSCs, which may alleviate systemic symptoms such as malaise or fatigue. MSC therapy has previously demonstrated its ability to dampen the overactivation of the immune system and cytokine storm in COVID-19 patients [59]. Given the established role of cytokines such as interleukin-1, interleukin-6, and tumor necrosis factor alpha in the pathophysiology of heart failure, it is plausible that the symptomatic relief associated with MSCs may be, at least partly, secondary to their immunomodulatory capacity [60]. Although the possibility of a placebo effect cannot be excluded, the consistency of MLHFQ improvement across numerous blinded trials suggests an underlying biological basis that warrants further exploration.

In terms of safety outcomes, the evidence in the current literature regarding stem cells’ effects on MACE mostly falls in line with our findings. Fu H. et al. concluded that although this intervention may significantly reduce heart failure rehospitalization rates, it does not influence cardiovascular death, rates of myocardial infarction (MI), and total death [53]. In contrast, Fisher et al. revealed that despite cell therapy being associated with a long-term reduction in the incidence of arrhythmias (RR = 0.42, 95% CI: 0.18 to 0.99) and the incidence of non-fatal myocardial infarction (RR = 0.38; 95% CI: 0.15 to 0.97), there was no evidence to support a significant decrease in HF rehospitalization rates (RR = 0.63; 95% CI: 0.36 to 1.09) [61]. Therefore, it can be deduced that the application of stem cell therapy to HF patients does not increase the risk of MACE but rather shows no difference when compared to placebos. While the current review indicates that the use of MSCs is not significantly associated with an increased risk of MACE, these results should be interpreted with caution, as the safety outcome was reported in only two studies (Studies 1 and 7), limiting the generalizability of the results.

Many factors underline the heterogeneity seen across studies regarding the efficacy and safety of MSC therapy for HF patients, with the primary factor being methodological variety. Brunskill SJ et al. compared the efficacies of intracoronary (IC) and intramyocardial (IM) delivery of stem cells, showing differences in LVEF (%) of −0.19% (95% CI: −0.24 to −0.15, *p* < 0.0001) and 5.85% (95% CI: 2.50 to 9.19, *p* = 0.0006), respectively, at 6-month follow-ups [62]. This shows that the direct injection of stem cells into the heart is more efficacious than delivering the cells via the coronaries. On the other hand, Ahmed ZT et al. assessed the dose–response relationship between stem cell therapy and heart failure [63].

In agreement with the PROMETHEUS and POSEIDON trials, research has demonstrated that efficacy is inversely correlated with the dose of stem cells. However, there has been no consensus on the dose–response relationship, as other trials, such as the TRIDENT trial, have reported improved LVEF (%) in patients who received higher doses of stem cells. Moreover, Gyongyosi et al. found that receiving IC BM-MSCs within two to seven days of experiencing an acute myocardial infarction improved LVEF (%) and decreased cardiac volumes significantly when compared to treatment within 24 h or beyond one week. This finding indicates that the timing of the intervention is critical [64]. Another source of heterogeneity is the source of MSCs. There are various origins of MSCs, such as the bone marrow, adipose tissue, and umbilical cord blood, which yield cells with diverse phenotypic and functional profiles and can, in turn, influence therapeutic outcomes [65].

### 4.3. Limitations

An important limitation of this review is the variability in follow-up time points. Some of the studies did not approach a one-year follow-up period; this is a limitation as it fails to capture the long-term effects and safety outcomes of the therapy in chronic heart failure patients [66]. One main limitation of this systematic review is the limited number of eligible and included studies. This may reflect the unwillingness of institutions to pursue stem cell therapy in the context of the unpromising results reported in studies from previous years. Notably, the small sample sizes in several of the included studies increase the chance of a type II error and reduce the statistical power of this meta-analysis. In addition, the studies included in this review demonstrated methodological heterogeneity, including with respect to the routes of stem cell administration and stem cell sources, and it must be considered that there is no unified consensus on which method or source is superior.

Heterogeneity was also seen with respect to the follow-up periods of the different studies, which ranged from 30 days to four years, with the majority of patients being followed up to 12 months. Moreover, the small sample sizes used in many of the studies increased the chance of a type-two error and may have reduced the external validity of the results, as observed for Studies 1, 4, 5, 8, and 12 [38,39,43,44,46]. Studies 3 and 12 [46,47] were single-arm, open-label, interventional studies; hence, there were no control groups from which to draw effective comparisons. Additionally, the same level of blinding was difficult to achieve in some studies because of the nature of the interventions and ethical considerations, such as in Studies 5 and 6 [44,45]. For instance, in Study 5 [44], the difference in resistance when administering single-cell injections as opposed to collagen–cell mixture injections prevented the physicians from being completely blinded. On the other hand, sham left-heart catheterization and bone marrow aspiration were not conducted in the control group in Study 6 [45] because of ethical implications, hindering the blinding process.

Furthermore, half of the studies included in this review utilized autologous stem cells (as shown in Studies 1, 3, 4, 6, 8, and 12 [38,39,43,45,46,47]), which, by their nature, introduce patient-to-patient variation with regard to the quality of the cells as well as the duration of cell cultures and expansion. This would be unlikely with the use of allogenic stem cells, which are more standardized in terms of quality. In terms of consistency, the methods of obtaining LVEF measurements differed, with some studies using echocardiography, such as Studies 2, 3, 4, 5, 6, 9, 10, 11, and 12 [37,40,41,42,43,44,45,46,47], while others employed cardiac MRI [19,38,39].

Meanwhile, certain limitations pertained to particular studies only. For instance, Study 7 [19] was terminated before 12 months because of futility; thus, it did not have enough power to detect differences in primary and secondary outcomes at 6 months. As for Studies 10 and 11 [40,41], the immunomodulatory and anti-inflammatory effects of APCs were determined to be optimal for patients afflicted with an ongoing inflammatory state, which was not the case for the majority of patients who had chronic LV dysfunction. This finding may explain their suboptimal outcomes. Lastly, the interval period of 30 days between randomization and index catheterization in Study 9 [42] allowed patients to drop out of the study, reducing the size of the study population.

This review highlights the importance of creating recommendations so that future research guidelines are set more clearly. First, it is recommended that patients are stratified based on key characteristics, such as age, disease comorbidities, or gender in order to better understand which patient population would benefit from MSC treatment. Additional suggestions are proposed in regard to the standardization of MSC regimens. It is recommended that there is a standard way of collecting, preparing, dosing and administering the treatment to avoid fragmented results due to confounding factors. Finally, because heart failure outcomes are better assessed in the long term, it is recommended that larger and multi-center trials with longer follow-up periods be conducted in this manner described above.

## 5. Conclusions

In conclusion, in HF management, MSC therapy is a novel approach to targeting myocardial wall stress reduction via Laplace’s law as opposed to targeting the conventional neurohormonal pathways. Although MSC therapy was determined to be safe, there was no statistically significant improvement in LVEF, scar burden, or left ventricular volume. However, there was a modest improvement in QoL. Therefore, despite the theoretical promise, this systematic review and meta-analysis failed to demonstrate the intervention’s applicability for HFrEF patients. Moreover, the variation in the findings of this review and others highlights the complexity of MSC therapy. This review showed that the hypothesis formulated was partially correct, considering that there was no significant effect of MSCs, as their administration resulted in a modest improvement in quality of life but not the other outcomes. This review also showed that there are no negative effects in regard to the proposed hypothesis. While MSC therapy remains an option for regenerative repair in heart failure, the current evidence does not support its efficacy in improving LVEF in HFrEF patients. The observed improvements in MLHFQ scores indicate the importance of requiring further investigations to view MSC therapy as experimental, and future trials should explore the methodological aspects of this review to optimize benefits for patients.

## Figures and Tables

**Figure 1 diseases-13-00136-f001:**
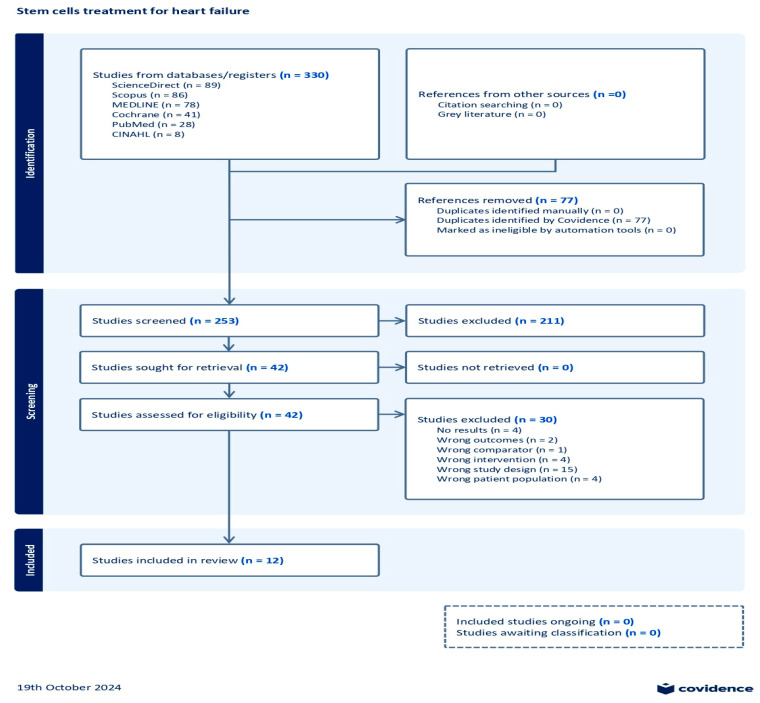
Prisma flow chart.

**Figure 2 diseases-13-00136-f002:**
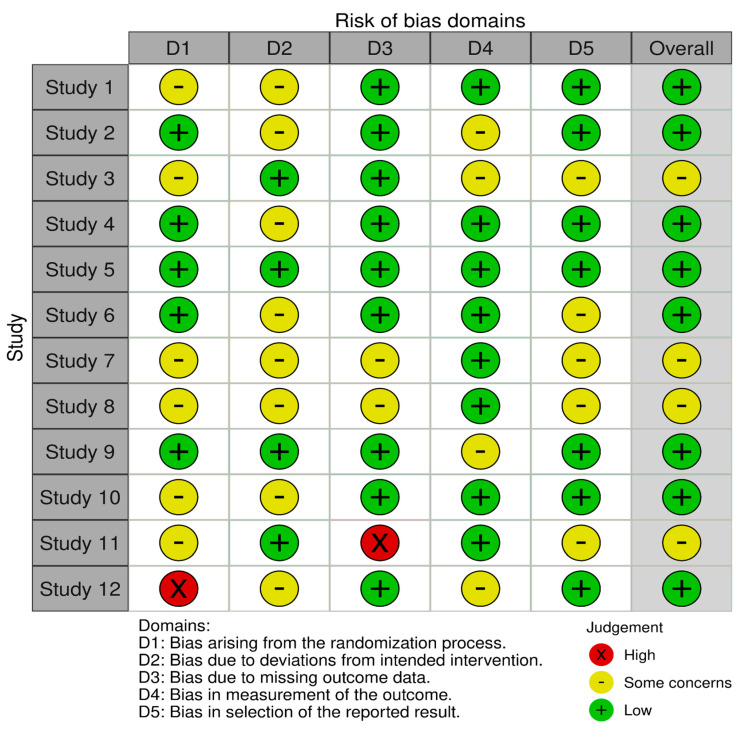
Risk-of-bias assessment.

**Table 1 diseases-13-00136-t001:** Search strategy for each database.

Database	Search Terms
PubMed	“Heart Failure”[Mesh] OR “Cardiac Failure”[tw] “Heart Failure, Systolic”[Mesh] OR “Systolic Heart Failure”[tw] OR “Adult Stem Cells”[Mesh] OR “Adult Stem Cell”[tw] OR “Stem Cell Research”[Mesh] OR “Stem cell therapy”[tw] OR “Stem cell”[tw] OR “Mesenchymal stem cell therapy”[tw] AND “Ventricular Function, Left”[Mesh]
Scopus	TITLE-ABS-KEY (heart AND failure OR cardiac AND failure OR systolic AND heart AND failure AND stem AND cell OR stem AND cell AND therapy OR mesenchymal AND stem AND cell AND left AND ventricular AND function) AND (INDEXTERMS (“clinical trials” OR “clinical trials as a topic” OR “randomized controlled trial” OR “Randomized Controlled Trials as Topic” OR “controlled clinical trial” OR “Controlled Clinical Trials” OR “random allocation” OR “Double-Blind Method” OR “Single-Blind Method” OR “Cross-Over Studies” OR “Placebos” OR “multicenter study” OR “double blind procedure” OR “single blind procedure” OR “crossover procedure” OR “clinical trial” OR “controlled study” OR “randomization” OR “placebo”)) AND (LIMIT-TO (OA, “all”)) AND (LIMIT-TO (DOCTYPE, “ar”)) AND (LIMIT-TO (PUBYEAR, 202) OR LIMIT-TO (PUBYEAR, 2022) OR LIMIT-TO (PUBYEAR, 2021) OR LIMIT-TO (PUBYEAR, 2020) OR LIMIT-TO (PUBYEAR, 2019) OR LIMIT-TO (PUBYEAR, 2018)) AND (LIMIT-TO (LANGUAGE, “English”))
EBSCO Medline	Heart Failure AND Stem cells AND left ventricular ejection fraction
EBSCO CINAHL	Heart Failure AND Stem cells AND left ventricular ejection fraction
Cochrane Library	Stem cell therapy AND Heart Failure AND Left Ventricular Function
ScienceDirect	Heart failure AND mesenchymal stem cell therapy AND left ventricular ejection fraction

**Table 2 diseases-13-00136-t002:** Overview of the studies selected for data extraction [19,37,38,39,40,41,42,43,44,45,46,47].

Study No.	Author, Year	Study Type	Number of Participants (MSC Group/Control Group)	Mean Age of Participants (MSC Group/Control Group)	Mean LVEF at Baseline (MSC Group/Control Group)	NYHA* Class III and IV (MSC Group/Control Group)	Method of Stem Cell Delivery	Control Group	Type of MSC	Patient Population	Follow-Up Time
1	Bolli, Roberto et al. (2021) [38]	Phase 2, randomized, placebo-controlled study with parallel assignment and quadruple masking	125	62.5	28.6 ± 6.1	80% NYHA class 2, 15% NYHA class 3	Transendocardial injection	Placebo	Autologous BM- MSCs	EF ≤ 40% on optimal GDMT	12 months
2	Bolli, Roberto et al. (2020) [37]	Phase I, randomized, placebo-controlled trial with quadruple masking	37	56.6 ± 11.8	33 ± 5.3%	84% NYHA class 2	Intramyocardial injections	Placebo	Allogeneic MSCs	EF ≤ 45% on optimal GDMT	12 months
3	Domae, Keitaro et al. (2021) [47]	Open-label single-arm interventional study	24	53.5 ± 2.6	-	33.3% NYHA class 2, 62.5% NYHA class 3	Stem cell patch implantation	-	Autologous skeletal stem cells	EF < 35% on optimal medical treatment	47.5 ± 4.3 months
4	Drabik, Leszek et al. (2022) [43]	Randomized control trial	10	60.8 ± 7.1	22.8 ± 4.5 CSCs, 27.8 ± 5.8 sham	-	Transendocardial injection	Sham procedure	Autologous BM- MSCs, cultured to form cardiopoietic stem cells	-	30 days
5	He, Xiaojun et al. (2020) [44]	Randomized double-blind clinical trial	50	62.6 ± 8.3	29.63% ± 6.03% in MSCs; 27.98% ± 7.39 in control	50% NYHA class 3 and 50% NYHA class 4 in MSCs; 58.3% NYHA class 3 and 41.7% NYHA class 4 in control	Intramyocardial injections while undergoing CABG**	CABG alone	Homologous, allogeneic UC-MSCs	LVEF ≤ 40%	12 months
6	Kim, Su Hyun et al. (2018) [45]	Randomized controlled trial	26	55.3 ± 8.6 in MSCs; 57.8 ± 8.9 in control	35.1 ± 4.5 in MSCs; 37.4 ± 1.7 in control	-	Intracoronary infusion after 30 days from PCI***	No sham procedure	Autologous BM-derived MSCs	LVEF ≤ 40% in patients who underwent PCI for a STEMI****	12 months
7	Makkar, Raj R et al. (2020) [19]	Multicenter randomized double-blind placebo-controlled trial	142	55 ± 11 in CDCs; 54 ± 10 in control	39.9 ± 6.6 in CDCs; 38.8 ± 8.2 in control	43.3% class 2 and 1.1% class 3 in CDCs; 52.3% class 2 and 11.4% class 3 in control	Intracoronary infusion	Intracoronary infusion of placebo	Allogeneic cardiosphere-derived cells (CAP-1002)	LVEF ≤ 40% with prior MI	12 months
8	Mathiasen, Anders B et al. (2020) [39]	Phase II, single-center, randomized, double-blind, placebo-controlled trial	60	66.1 ± 7.7 in MSCs; 64.2 ± 10.6 in control	-	27.5% class 2 and 72.5% class 3 in MSCs; 25.0% class 2 and 75.0% class 3 in control	intramyocardial injections	Intramyocardial injections of placebo (saline)	Autologous BM-MSCs	Ischemic HF with LVEF ≤ 45%	4 years
9	Perin, Emerson C et al. (2023) [42]	Phase 3, multinational, randomized, double-blind, sham-controlled clinical trial	565	62.7 ± 10.9 in MSCs; 62.7 ± 10.9 in control	28.6 ± 6.7 in MSCs; 28.6 ± 6.9 in control	38.2% class 2 and 61.8 class 3 in MSCs; 36.9% class 2 and 63.1% class 4 in control	Transendocardial injections	Sham control procedure	Allogeneic bone-marrow-derived mesenchymal precursor cells	LVEF ≤ 40%	12 months
10	Qayyum, Abbas Ali et al. (2023) [40]	Multi-center, double-blind, placebo-controlled phase II study	81	67.0 ± 9.0 in ASCs; 66.6 ± 8.1 in control	34.2 ± 7.9 in ASCs; 31.4 ± 7.2 in control	Mean NYHA class 2.2 ± 0.4 in ASCs and 2.3 ± 0.4 in control	Intramyocardial injections	Placebo (saline) injection	Allogenic adipose-tissue-derived mesenchymal Stromal cells	LVEF ≤ 45%	12 months
11	Qayyum, Abbas Ali et al. (2023) [41]	Multicenter, double-blind, placebo-controlled phase II trial	133	66.4 ± 8.1 in ASCs; 64.0 ± 8.8 in control	31.6 ± 7.2 in ASCs; 32.0 ± 8.9 in control	68.9% class 2 and 31.1% class 3 in ASCs; 69.8% class 2 and 30.2% class 3 in control	Intramyocardial injections	Placebo (saline) injection	Allogenic adipose-tissue-derived mesenchymal stromal cells	LVEF ≤ 45%	12 months
12	Sato, Yukihito et al. (2021) [46]	Preliminary, interventional, single-arm, open-label study	5	70.0 ± 4.0	29.2 ± 7.6	-	Intravenous infusion		Autologous BM- MSCs	LVEF ≤ 45%	6 months

## Data Availability

Data sharing is not applicable.

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
