# Peer review of "Cardiac Repair and Clinical Outcomes of Stem Cell Therapy in Heart Failure: A Systematic Review and Meta-Analysis"

_diseases, 2025, doi:10.3390/diseases13050136_

Round 1
Reviewer 1 Report
Comments and Suggestions for Authors
The presented meta-analysis is interesting, well written, good English, clear conclusions. Major/minor remarks:
- There are too many abbreviations, full names should be given in the Figures as the figures often are read independently of the text; also please give full names when the abbreviations appear in the main text (abstract and text are independent parts of the paper)
- The quality of the charts must be improved; font size enlarged, fig 13 replaced with a better quality.

Author Response
Comment 1: There are too many abbreviations, full names should be given in the Figures as the figures often are read independently of the text; also please give full names when the abbreviations appear in the main text (abstract and text are independent parts of the paper)
We have addressed this part by giving full name of the abbreviations in text and figures and also gave full names to abbreviations in abstract and remaining paper independently.
Comment 2: The quality of the charts must be improved; font size enlarged, fig 13 replaced with a better quality.
The charts are optimized for quality. Figure 13 is being requested to take from biostatistician and will be replace in galley proof.
Reviewer 2 Report
Comments and Suggestions for Authors
Manuscript Evaluation Report
Title "Repairing the Heart, Stem by Stem: The Role of Mesenchymal Stem Cell Therapy in Heart Failure: A Systematic Review and Meta-analysis"
General Comments
The manuscript presents a systematic review and meta-analysis on the therapeutic use of mesenchymal stem cells (MSCs) in patients with heart failure with reduced ejection fraction (HFrEF). The study is well structured, follows the required general guidelines, and addresses a relevant and current issue in the scientific community. However, it requires methodological and editorial improvements to ensure its suitability for publication.
Detailed Comments
Title
The title is clear and engaging, but could be more concise avoiding the redundancy of the term "stem."
Abstract
Well structured; however, it needs to clearly specify the hypothesis or objective raised. Briefly include how the literature was discriminated against (clear and specific inclusion/exclusion criteria).
Introduction
It contextualises well the global problem of heart failure. It is recommended to synthesise and specify the current research context in MSCs for HF more clearly. The rationale for the review is adequate but could be more incisive on specific knowledge gaps.
Materials and Methods
Well developed according to PRISMA. It is necessary to clearly explain the specific criteria used to discriminate studies (e.g. minimum methodological quality required).
Results
Clear presentation of the results of the meta-analysis, although the tables and figures require adjustments to meet the required visual quality. There is a lack of figures and tables that effectively summarise the literature reviewed according to the Diseases journal guideline.
Discussion
Good critical analysis and comparison with other reviews. It is suggested to discuss in more depth the specific mechanisms by which MSCs could fail to improve LVEF, exploring the biological implications. It is recommended that future research guidelines be set out more clearly, especially in view of the limitations highlighted.
Conclusions
Adequately synthesise the results of the study. It could be further refined to respond directly to the specific hypothesis or objective formulated.
References
Wide and adequate. Verify format in accordance with the editorial standards of Diseases (MDPI).
Recommendations
Clearly, specify in the summary and introduction the central hypothesis or objective. Improve the visual quality and detail of figures and tables. To deepen the explanation of the cellular mechanisms involved in the discussion. Indicate precisely the methodological criteria used to select and evaluate the studies.
Strengths and Weaknesses
Strengths
A topic of high clinical and scientific relevance. Good overall systematic review methodology. Extensive search and appropriate use of recent literature.
Weaknesses
Lack of specificity in the explicit formulation of hypotheses and objectives. Limitations in the visual quality and clarity of the tables and figures. Insufficient discussion of the underlying mechanisms and specific knowledge gaps.
Publishability
The manuscript has good scientific quality; however, it requires minor modifications before it is considered suitable for publication in the journal Diseases. After making the improvements suggested in this report, especially in objective accuracy, visual quality of figures, and depth in discussed mechanisms, the article will be ready for publication.
Reviewer 2 Report
Comments and authors responses:
Title
The title is clear and engaging, but could be more concise avoiding the redundancy of the term "stem."
Reviewer response: Thank you for your insightful comment. We have addressed by changing the title to “Harnessing cellular repair: The role of Mesenchymal Stem Cell Therapy in Heart Failure: A Systematic review and Meta-analysis”
Abstract
Well structured; however, it needs to clearly specify the hypothesis or objective raised. Briefly include how the literature was discriminated against (clear and specific inclusion/exclusion criteria).
Reviewer response: We would like to thank you for your thoughtful comment. This was addressed by adding a statement regarding the hypothesis “It was hypothesized that the administration of MSCs is both safe and efficacious in HFrEF patients” to the introduction section of the abstract and the manuscript. We clarified the inclusion criteria by rewording “Eligible studies” to “Included studies”. We also added a sentence in the methods section to specify the exclusion criteria by stating “Exclusion criteria included observational studies, pediatric patients, and individuals with heart failure with preserved ejection fraction (HFpEF).”
Introduction
It contextualises well the global problem of heart failure. It is recommended to synthesise and specify the current research context in MSCs for HF more clearly. The rationale for the review is adequate but could be more incisive on specific knowledge gaps.
Reviewer Response : Thank you for the insightful comment. We have addressed this by incorporating changes into the introduction to address the concerns. Firstly, we addressed the concern on the rationale for the review by adding it to section 1.4 (Importance of this review) seen as “Existing systematic reviews either include outdated studies or fail to solely focus on HFrEF patients, thereby limiting their clinical relevance. Consequently, this systematic review and meta-analysis bridges this gap by recruiting the most recent clinical trials to cohesively assess MSCs therapy in HFrEF patients through an updated and focused approach.”
Materials and Methods
Well developed according to PRISMA. It is necessary to clearly explain the specific criteria used to discriminate studies (e.g. minimum methodological quality required).
Author response : We acknowledge and thank you for this insight. Regarding this we have added a concise paragraph explaining the minimum methodological quality required when discriminating against allocated studies. This can be seen under section 2.2 after inclusion and exclusions paragraphs stated as “Regarding minimal methodological quality requirement, only studies that demonstrated low to moderate risk of bias based on the Cochrane Risk of Bias (RoB 2) tool were included. Additionally, all the included studies had to 1)”
Results
Clear presentation of the results of the meta-analysis, although the tables and figures require adjustments to meet the required visual quality. There is a lack of figures and tables that effectively summarise the literature reviewed according to the Diseases journal guideline.
Author response: Thank you for the thoughtful comment. We would like to respond and reassure you that we have abided by the Diseases journal guidelines in multiple aspects. Firstly, figures, and tables were in close proximity of their first citation for instance seen in Table 2 where we give an overview of studies selected for data extraction right under the section of study characteristics. Content was appropriate in English language with convenient symbols. Tables included explanatory headings; font was above 8 even in larger tables. As for the lack of figures, we would like to emphasize that we incorporated as many figures and tables in various ways , such as Figure 1 PRISMA flow diagram, Table 1 “Search strategy”, Table 2 “Overview of studies selected for Data extraction”, Figure 2 “Risk of bias assessment ”, in addition to the remaining of the figures including all the forest plots for the meta-analysis of the review. We believe this covers the major guidelines of the journal and is sufficient in regard to it.
Discussion
Good critical analysis and comparison with other reviews. It is suggested to discuss in more depth the specific mechanisms by which MSCs could fail to improve LVEF, exploring the biological implications. It is recommended that future research guidelines be set out more clearly, especially in view of the limitations highlighted.
Author response: Thank you for your valuable suggestion. Regarding the first suggestion, we have further explored the biological mechanisms through which MSCs failed to improve LVEF in section 4.2. Some of the reasons highlighted included low cellular retention, poor cellular preconditioning, and the transient nature of the paracrine effects associated with MSCs. As for the recommended guidelines, we have taken this comment into consideration and enhanced our discussion section by highlighting important suggestions involving standardization methods, increasing size of the clinical trials conducted and expanding it to multi-center studies to better address long term outcomes of MSCs, and patient stratification prior to study conduction. These changes are seen in detail under section 4.5 “Limitations” in the final paragraph. Once again, we appreciate the insightful suggestions.
Conclusions
Adequately synthesize the results of the study. It could be further refined to respond directly to the specific hypothesis or objective formulated.
Author Response : Thank you for the valuable feedback. To address this the following changes were made in the revised manuscript, whereby a statement was added to directly answer the specific hypothesis formulated as seen in conclusion section.
References
Wide and adequate. Verify format in accordance with the editorial standards of Diseases (MDPI).
Recommendations
Clearly, specify in the summary and introduction the central hypothesis or objective. Improve the visual quality and detail of figures and tables. To deepen the explanation of the cellular mechanisms involved in the discussion. Indicate precisely the methodological criteria used to select and evaluate the studies.
Strengths and Weaknesses
Strengths
A topic of high clinical and scientific relevance. Good overall systematic review methodology. Extensive search and appropriate use of recent literature.
Weaknesses
Lack of specificity in the explicit formulation of hypotheses and objectives. Limitations in the visual quality and clarity of the tables and figures. Insufficient discussion of the underlying mechanisms and specific knowledge gaps.
Author response: We have tried to address the study weaknesses as identified by adding information about the hypothesis and study objective. We have also tried to improve the quality of the tables and figures optimally. We have further enhanced the discussion in each area and covered literature gaps.
Publishability
The manuscript has good scientific quality; however, it requires minor modifications before it is considered suitable for publication in the journal Diseases. After making the improvements suggested in this report, especially in objective accuracy, visual quality of figures, and depth in discussed mechanisms, the article will be ready for publication.
Author response: We have tried to address all the recommendations. We are hopeful it will satisfy the reviewer comments.

Reviewer 3 Report
Comments and Suggestions for Authors
I appreciate the authors’ effort in compiling a systematic review and meta-analysis on the use of mesenchymal stem cell (MSC) therapy in heart failure with reduced ejection fraction (HFrEF). The topic is clinically relevant and timely, given the increasing interest in regenerative strategies beyond guideline-directed medical therapy (GDMT). The manuscript is generally well-structured, with appropriate adherence to PRISMA methodology. However, there are several key concerns regarding scope, analytical depth, interpretation of results, and presentation that need to be addressed before the manuscript is suitable for publication.
Major Points:
1. The title is slightly misleading. It suggests a more conclusive or optimistic outcome than the data support. The findings mostly show no statistically significant benefit in key cardiac parameters. Consider softening the title to better reflect the actual conclusions.
2. Although you’ve restricted inclusion to trials since 2018, the conclusions are consistent with previous reviews. What new insights does your review add to the existing body of literature? This should be clearly articulated in the introduction and discussion.
3. The included studies used autologous vs. allogeneic cells, different sources (BM, adipose, umbilical cord), and various routes of delivery (intramyocardial, intracoronary, IV). These factors are likely to influence outcomes but are not adequately analyzed or discussed as sources of variability.
4. The statistically significant improvement in MLHFQ scores is worth highlighting more thoroughly. What might explain this subjective benefit in the absence of objective improvement? A brief exploration of the potential placebo effect or paracrine signaling would be helpful.
5. Use of only two studies for safety outcome (MACE) is a limitation. This is underplayed in the discussion. With such limited data, it’s hard to draw meaningful conclusions on safety. Please be more cautious in how this is interpreted.
6. Search strategy appears thorough, but the decision to exclude non-English studies may have introduced bias. A justification for this language restriction should be included, as stem cell studies are conducted worldwide.
7. Meta-analysis may be underpowered. Several included studies had small sample sizes. A comment on the risk of type II error and its implications for interpreting null findings should be added.
8. Forest plots are included but could be better labeled (e.g., sample sizes, MSC type, study name on the axis). Additionally, summary tables for each outcome would help readers quickly digest the main findings.
9. The discussion lacks a clear “take-home message.” A paragraph should be added near the end of the discussion summarizing what clinicians, researchers, and policy makers should conclude from these findings.
10. PRISMA checklist and risk of bias summary are mentioned but not included in detail. These are critical for transparency in systematic reviews and should be added as supplementary material.
Minor Points:
11. There are numerous minor grammatical issues and awkward phrasings. A thorough language edit is recommended.
12. Please be consistent with abbreviations (e.g., HFrEF, MSCs).
13. The conclusion could be more concise. The current version repeats earlier points without adding much.
Comments on the Quality of English LanguageThere are several awkward phrasings, repetitive transitions (e.g., henceforth), and occasional wordiness that affect the overall readability.
Author Response
Comments and Suggestions for Authors
I appreciate the authors’ effort in compiling a systematic review and meta-analysis on the use of mesenchymal stem cell (MSC) therapy in heart failure with reduced ejection fraction (HFrEF). The topic is clinically relevant and timely, given the increasing interest in regenerative strategies beyond guideline-directed medical therapy (GDMT). The manuscript is generally well-structured, with appropriate adherence to PRISMA methodology. However, there are several key concerns regarding scope, analytical depth, interpretation of results, and presentation that need to be addressed before the manuscript is suitable for publication.
Reviewer comments and Author responses:
Major Points:
- The title is slightly misleading. It suggests a more conclusive or optimistic outcome than the data support. The findings mostly show no statistically significant benefit in key cardiac parameters. Consider softening the title to better reflect the actual conclusions.
Author response: We thank you sincerely for the insightful comment, for this we have changed the title for it to not be misleading towards a positive and conclusive outcome. We hope this update addresses this concern.
- Although you’ve restricted inclusion to trials since 2018, the conclusions are consistent with previous reviews. What new insights does your review add to the existing body of literature? This should be clearly articulated in the introduction and discussion.
Author response: Dear reviewer thank you for this insightful comment. To address this point, we have pointed out that the review exclusively examines clinical trials from 2018 onwards to reflect the most current advancements in stem cell treatment, including refinements in cell sourcing, delivery methods, and standardized outcome reporting. Additionally, our approach utilizes the most updated tool to assess the risk of bias (RoB 2). We focus on clinically validated QoL metrics such as the MLFHQ questionnaire. Although our review confirmed prior findings regarding limited improvement in LVEF, we noted a consistent trend towards improving patient-reported outcomes, suggesting a possible paradigm shift in how MSC efficacy should be evaluated, moving beyond purely structural endpoints. This can be found in section 2.2, the first paragraph.
- The included studies used autologous vs. allogeneic cells, different sources (BM, adipose, umbilical cord), and various routes of delivery (intramyocardial, intracoronary, IV). These factors are likely to influence outcomes but are not adequately analyzed or discussed as sources of variability.
Author response : Thank you for your feedback, we have addressed this by discussing heterogeneity factors underlying the chosen studies in section 4.2. This was also explained in the limitations sections.
- The statistically significant improvement in MLHFQ scores is worth highlighting more thoroughly. What might explain this subjective benefit in the absence of objective improvement? A brief exploration of the potential placebo effect or paracrine signaling would be helpful.
Author response: Thank you for your insightful comments. The cause behind MLHFQ improvement in the absence of improvement within other variables remains unknown. Potential causes could include immunomodulation which may reduce systemic symptoms such as malaise or fatigue. Additionally, paracrine signalling associated with MSCs may attenuate inflammation and oxidative stress, further contributing towards improvement in symptoms like fatigue. It is also worth stating that the possibility of a placebo effect cannot be excluded. However, the consistency of MLHFQ improvement across numerous blinded trials suggests an underlying biological basis that warrants further exploration. With that being said, an improvement in MLHFQ scores, despite an unchanged LVEF or structural improvement, accentuates the importance of patient-centered outcomes, as it can offer a more complete picture of clinical utility. This can be found in section 4.2 , paragraph 6.
- Use of only two studies for safety outcome (MACE) is a limitation. This is underplayed in the discussion. With such limited data, it’s hard to draw meaningful conclusions on safety. Please be more cautious in how this is interpreted.
Author response : Thank you for your insightful comment, we appreciate the your observation regarding the limited number of studies reporting on safety outcomes. While it is true that only two studies explicitly reported on MACE, both were randomized controlled trials with adequate follow-up durations and well-documented adverse event monitoring protocols. Their findings consistently showed no increase in serious adverse events associated with MSC therapy, which aligns with the broader body of literature suggesting a favorable safety profile for MSCs in cardiac and non-cardiac applications.
That said, we acknowledge that firm conclusions regarding safety cannot be drawn from such a small sample. In response to this comment, we have added a sentence in the discussion section explicitly stating that the safety outcomes should be interpreted with caution due to the limited number of studies contributing data to this analysis. This can be seen in section 4.2, paragraph 7.
- Search strategy appears thorough, but the decision to exclude non-English studies may have introduced bias. A justification for this language restriction should be included, as stem cell studies are conducted worldwide.
Author response : Thank you for highlighting the potential for language bias. The decision to exclude non-English studies was made to ensure the accuracy of data extraction, interpretation of methodology, and risk of bias assessment, particularly given the technical nature of clinical trials and the nuanced reporting of outcomes. While we acknowledge that this approach may limit global generalizability, it was deemed necessary to maintain methodological rigor and consistency.
In response to this comment, we have added a sentence in the methods section 2.2 acknowledging this limitation and justifying the rationale behind the English-language restriction.
- Meta-analysis may be underpowered. Several included studies had small sample sizes. A comment on the risk of type II error and its implications for interpreting null findings should be added.
Author response : Thank you for the point regarding the potential for limited statistical power due to small sample sizes in several of the included trials. While our analysis followed standard meta-analytic techniques, we recognize that some included studies had relatively small cohorts, which increases the risk of a Type II error—failing to detect a true effect where one may exist.
To address this, we have added a note in the limitations section of the discussion, acknowledging the possibility of underpowering and the need for larger, multicenter trials to more definitively assess MSC efficacy in HFrEF.
- Forest plots are included but could be better labeled (e.g., sample sizes, MSC type, study name on the axis). Additionally, summary tables for each outcome would help readers quickly digest the main findings.
- The discussion lacks a clear “take-home message.” A paragraph should be added near the end of the discussion summarizing what clinicians, researchers, and policy makers should conclude from these findings.
Author response : Thank you for your insight. We added this paragraph accordingly in the conclusion paragraph as the final statement.
- PRISMA checklist and risk of bias summary are mentioned but not included in detail. These are critical for transparency in systematic reviews and should be added as supplementary material.
Author response : PRISMA checklist and risk of bias are addressed appropriately to cover the subject requirement. We found it in accordance with other systematic reviews published in same domain.
Minor Points:
- There are numerous minor grammatical issues and awkward phrasings. A thorough language edit is recommended.
- Please be consistent with abbreviations (e.g., HFrEF, MSCs).
- The conclusion could be more concise. The current version repeats earlier points without adding much.
Comments on the Quality of English Language
There are several awkward phrasings, repetitive transitions (e.g., henceforth), and occasional wordiness that affect the overall readability.
Author response : The minor points regarding the grammatical and phrasing issues have also been comprehensively revised. We are thankful to the MDPI author services for providing us great support in addressing language and subject-related issues effectively.
We have made the conclusion more overarching and also avoided repetition as guided.

Round 2
Reviewer 3 Report
Comments and Suggestions for Authors
This revised manuscript has been improved by addressing the reviewer’s comments. However, the legend for Figure 8 appears to be missing, please add it. Additionally, all figure legends should be revised to include more detailed information.
Author Response
Thank you for your valuable insight during the review process. We have updated the figure 8 with legends and all figures are reassessed for detailed information.